# Amphipathic Small Molecule AZT Compound Displays Potent Inhibitory Effects in Cancer Cell Proliferation

**DOI:** 10.3390/pharmaceutics13122071

**Published:** 2021-12-03

**Authors:** Pethaiah Gunasekaran, Ho Jin Han, Jung hoon Choi, Eun Kyoung Ryu, Nam Yeong Park, Geul Bang, Yeo Kyung La, Sunghyun Park, Kyubin Hwang, Hak Nam Kim, Mi-Hyun Kim, Young Ho Jeon, Nak-Kyun Soung, Jeong Kyu Bang

**Affiliations:** 1Division of Magnetic Resonance, Korea Basic Science Institute (KBSI), Ochang, Cheongju 28119, Korea; gunaharaks@kbsi.re.kr (P.G.); ekryu@kbsi.re.kr (E.K.R.); pny940109@kbsi.re.kr (N.Y.P.); lyk1234@kbsi.re.kr (Y.K.L.); sunghyun0722@kbsi.re.kr (S.P.); kyubin_hwang@dgu.ac.kr (K.H.); haknam81@kbsi.re.kr (H.N.K.); 2Dandicure Inc., Ochang, Cheongju 28119, Korea; 3Anticancer Agent Research Center, Korea Research Institute of Bioscience and Biotechnology, Ochang, Cheongju 28116, Korea; hjhan@kribb.re.kr; 4Biomedical Omics Group, Korea Basic Science Institute, Ochang, Cheongju 28119, Korea; jhchoi19@kbsi.re.kr (J.h.C.); bangree@kbsi.re.kr (G.B.); 5Department of Bio-Analytical Science, University of Science & Technology, Daejeon 34113, Korea; 6Department of Internal Medicine, Pusan National University School of Medicine and Biomedical Research Institute, Pusan National University Hospital, Busan 49241, Korea; mihyunkim@pusan.ac.kr; 7College of Pharmacy, Korea University, 2511 Sejong-ro, Sejong 30019, Korea

**Keywords:** anticancer, oncosis, necrosis, drug discovery, small molecule

## Abstract

Cancer has been identified as a leading cause of death worldwide, and the increasing number of cancer cases threatens to shorten the average life expectancy of people. Recently, we reported a 3-azido-3-deoxythymidine (AZT)-based amphipathic small molecule, ADG-2e that revealed a notable potency against tumor metastasis. To evaluate the anticancer potential of ADG-2e, we assessed its anticancer potency in vitro and in vivo. Anticancer screening of ADG-2e against cervical cancer cells, HeLa CCL2, and BT549 mammary gland ductal carcinoma showed significant inhibition of cancer cell proliferation. Furthermore, mechanistic investigations revealed that cancer cell death presumably proceeded through an oncosis mechanistic pathway because ADG-2e treated cells showed severe damage on the plasma membrane, a loss of membrane integrity, and leakage of *α*-tubulin and *β*-actin. Finally, evaluation of the antitumorigenic potential of ADG-2e in mouse xenograft models revealed that this compound potentially inhibits cancer cell proliferation. Collectively, these findings suggest that ADG-2e can evolve as an anticancer agent, which may represent a model for nucleoside-based small molecule anticancer drug discovery.

## 1. Introduction

Cancer and its consequences continue to pose a threat to humanity by causing serious damages to socio-economic factors. According to WHO reports, cancer is the leading cause of death worldwide, with approximately 19.3 million new cases and 10.0 million deaths in 2020. Additionally, the projected global cancer occurrence in 2040 is approximately 28.4 million, an increase of 47% [1]. Human cancer involves complex processes, where cellular and molecular alterations are stimulated by various endogenous and exogenous factors that trigger uncontrollable cell growth by abnormal cell divisions [2,3,4]. Cancer requires an arsenal of treatment methodologies. Early detection, surgery, radiation therapy, and chemotherapy are critical approaches that have been most effective in treating tumors to reduce cancer mortality. It is imperative to note that the prevalent application of small-molecule chemotherapeutics is an inevitable choice because of their remarkable potency against a wide range of tumors and their ability to penetrate solid tumors [5,6]. Therefore, identification of highly effective small molecule anticancer drugs for chemotherapy is an attractive strategy for anti-cancer drug discovery, despite being highly challenging [7].

The cancer cell surface is enriched with negatively charged phospholipids, phosphatidylserine (PS) that is susceptible to electrostatic interactions with cationic groups in anti-cancer drugs [8,9]. In contrast, in normal cells, the negatively charged phospholipids projected in the inner leaflet, and the outer leaflet of the plasma membrane are rich in neutral phospholipids, such as phosphatidylcholine (PC) and sphingomyelin (SM). The reversal of phospholipid patterns in tumour cells is usually induced by the generation of inflammatory cytokines, oxidative stress, acidity, and thrombin [10].

Cationic peptide antibacterials (CPAs) are amphipathic peptides that comprise balanced portions of cationic and hydrophobic residues [9,11]. Given that the membrane phenotype of bacteria also constitutes negatively charged phospholipids and lipopolysaccharides, numerous naturally occurring CPAs have been found to inhibit cancer [12,13,14]. Consequently, several naturally occurring CPAs have progressed to clinical trials as anticancer therapeutics. For instance, LL37 advanced to phase I and II clinical trials by showing promising efficacy against melanoma, colon, and gastric cancers [15]. In addition, LTX-315, a derivative of bovine lactoferricin, showed prominent potency against several drug-resistant cancer cell lines and advanced to phase II clinical trials [16,17]. Similarly, clinical phase II results of dusquetide (SGX942), an innate defense regulator (IDR), revealed that it inhibited severe oral mucositis (SOM) in patients with head and neck cancer [18]. Buforin IIb showed cytotoxicity against 62 cancer cell lines, including leukemia, breast, prostate, and colon cancer [19]. Melittin was found to be effective against skin cancers [20]. In addition, human neutrophil peptide-1(HNP-1) [21], aurein 1.2 [22], and pleuricidin [14,17] are other CPAs that show promising effects against various drug-resistant cancer cell lines.

Although many CPAs have proven to be remarkable anticancer peptides (ACPs), their transformation into therapeutics is hampered by potential disadvantages, including poor stability, proteolytic degradation, short plasma half-life, and low bioavailability [23,24]. In addition, to perform a structure-activity relationship study (SAR), synthesis of long-sequenced peptides involves huge manufacturing costs due to complex synthetic routes to locate the desired functional groups at appropriate positions. The production cost is reported to be more than 10-fold higher than the manufacturing cost of small molecules [25]. Thus, designing amphipathic small molecules that mimic the structural features of CPAs is an effective strategy [24,26].

To accomplish this, we envisioned 3′-azido-3′-deoxythymidine (AZT), which offers synthetic feasibility to incorporate desired functional groups to attain amphipathicity. Furthermore, in recent years, AZT-based therapeutics have drawn significant attention because of their proven medicinal importance as HIV drugs [27], antitumor effects [28], antioxidant [29], and AZT prodrugs for anticancer and antiviral activities [30,31]. In particular, AZTs have a wide range of applications as anticancer agents that inhibit several tumor cell lines, including colon [32], breast [33], bladder [29,34], and esophageal cancers [35]. 

Recently, we reported a series of amphipathic 3′-azido-3′-deoxythymidine (AZT) derivatives containing amine or guanidine sidechains to mimic lysine and arginine, respectively, which were balanced by various hydrophobic residues. Through intensive SAR, we identified ADG-2e as a potent CPA that showed notable antimetastatic activity [36]. It is pertinent to note that metastasis is the prime factor that increases cancer mortality by causing failures in cancer treatment [37,38]. Even though the present clinical application of anticancer drug numbers greater than 200, most of them did not inhibit cancer metastasis [39,40], prompting an urgent need to develop new therapeutic modalities that treat cancer and metastasis simultaneously. In addition, inspired by the clinical use and preclinical development of nucleosides in the treatment of cancer [41,42], and to explore the therapeutic potential of ADG-2e, we evaluated its anticancer potential in vitro and in vivo. The mechanism of action on cancer cells was investigated by studying morphological changes in the cell membranes.

## 2. Materials and Methods

### 2.1. Chemistry

ADG-2 was synthesized according to the procedure [36], and the experimental section is provided in the Appendix A. As described previously [43], all reactions were performed under argon atmosphere in flame-dried glassware using dry solvents, unless otherwise noted. Anhydrous organic solvents of purity greater than 99.9% were purchased from Aldrich and used directly in the reaction. All reagents and few starting materials were purchased from commercial chemical suppliers, including Sigma-Aldrich, TCI, and Across Organics and used as received. Analytical thin-layer chromatography (TLC) was performed on Merck aluminum sheets with silica gel 60 F254 using 0.25 mm plates and was visualized by ultraviolet light, staining with KMnO4 and ninhydrin. Column chromatography purification was performed on Merck silica gel 60 (70–230 mesh or 230–400 mesh). Bruker DRX-500 and DRX-800 NMR spectrometers were used to record 1H and 13C NMR spectra. NMR chemical shifts (δ) are denoted in parts per million (ppm) and coupling constants (J) are given in hertz (Hz). MALDI-TOF mass was recorded using a Shimadzu mass spectrometer. 

### 2.2. Cell Culture and Cell Viability Assay

HeLa, BT549 and MRC5 were purchased from the American Type Culture Collection (ATCC, Manassas, VA, USA). CCD34SK and CCD986SK cells were obtained from the Bio Evaluation Center of KRIBB. HeLa CCL2, BT549, and MRC5 cells were grown in DMEM, and CCD34SK cells were grown in EMEM medium. CCD986SK growth was observed in the IMDM media. All media contained penicillin/streptomycin (Thermo Fisher Scientific, Waltham, MA, USA) and 10% fetal bovine serum (Gibco, Logan, UT, USA). The cells were incubated at 37 °C with 5% CO_2_. HeLa CCL2 cells (4 × 10^3^ cells/well) were seeded in 96-well cell culture plates and incubated for 18 h. Cells were treated with various concentrations of ADG-2e. All experiments were performed in triplicate. After 24 h of additional incubation, 10 μL of MTT solution (Daeil Lab service, Daejeon, Korea) was directly added and incubated at 37 °C for 2 h. The absorbance was measured at 450 nm using a SPARK 10M (TECAN, Männedorf, Switzerland) and IC_50_ was measured using GraphPad Prism 6.0 program (San Diego, CA, USA)

### 2.3. Live and Dead Assay

HeLa CCL2 cells were seeded in 96-well plates and incubated for 18 h. Then, 0, 25, and 50 µM ADG-2e were added and incubated for an additional 24 h. 2 µM of Calcein-AM (Live-cell; Green) and 4 µM EthD-1 (Dead cell; Red) were treated on each well for 30 min and incubated at 37 °C. DNA was stained by Hoechst 33342 (Blue; Sigma-Aldrich, St. Louis, MO, USA). Images were captured using a fluorescence microscope (Carl Zeiss). 

### 2.4. FACS Analysis

HeLa CCL2 cells were seeded in 12-well plates for 18 h. Cells were treated with 25 μM ADG-2e. After incubation for an additional 24 h, cells were harvested and stained with propidium iodide (PI) or anti-Annexin V-APC (eBioscience TM Annexin V Apoptosis Detection Kit APC, Invitrogen, Carlsbad, CA, USA) for 30 min. Cell death was measured by flow cytometry (CytoFLEX; Beckman Coulter, Miami, FL, USA). Wavelengths of 650/660 nm were used for laser excitation and emission, respectively, for APC detection. Wavelengths of 535/617 nm were used as excitation and emission wavelengths, respectively, for PI detection. The results were analyzed using FlowJo software (BD Biosciences, Franklin Lakes, NJ, USA).

### 2.5. Live Image Assay

HeLa cells were seeded in Lab-Tek II 2 chamber cover glass wells (Nunc, Rochester, NY, USA) and incubated at 37 °C for 18 h. Then, they were placed on the stage of the Zeiss microscope (Carl Zeiss Microimage Inc., Thornwood, NY, USA) with ZEN software and inverted microscope equipped with an environmental chamber (Precision Plastics, Beltsville, MD, USA) to provide temperature, humidity and CO_2_ control. The cells were treated with 0, 25, or 50 µM ADG-2e. Time-lapse images were captured every 2 min using a differential interference contrast (DIC) microscope.

### 2.6. Tumor Xenograft Analysis and Immunohistochemistry

All animal studies were conducted in accordance with relevant guidelines and regulations approved by the Institutional Animal Care and Use Committee of KRIBB (No.: KRIBB-AEC-20258, approval date, 23 October 2020). Male BALB/c nude mice (six weeks) were purchased from Daehan Biolink. 3 × 10^5^ HeLa cells/mouse were inoculated subcutaneously into mice. When tumor volumes reached 100 mm^3^, the mice were divided into two groups. Each group consisted of six mice, and three times a week, IP injections of 30 mg/kg of ADG-2e or vehicle (20% DMA (*N*, *N*′-dimethylacetamide), 20% Kolliphor, 60% HPBCD(2-hydroxypropyl)-β-cyclodextrin) were administered until the tumor volume reached 1000 mm^3^. Every three days interval the volume of tumors was measured. The day after the final injection, the mice were sacrificed, and images were captured. The tumor tissues were harvested and fixed in 4% paraformaldehyde overnight at 4 °C. For immunohistochemistry, the sections were deparaffinized and blocked overnight with 5% BSA. Sections were incubated with primary antibodies, such as anti-cleaved caspase 3 and *α*-tubulin antibodies, overnight at 4 °C. After washing with DPBS three times, the sections were incubated with secondary antibodies for 1 h. DNA was stained with Hoechst 33342 (Sigma-Aldrich).

### 2.7. Mouse Behavior Analysis in an Open Field Cage

The apparatus was similar to that used by Reyes-Mendez et al. [44], which consisted of a dark (black) acrylic enclosure, with a 30  ×  30 cm floor area and a 40 cm height surrounding wall. We investigated the mouse in an open field cage for 7 min, which consisted of squares. The video data were analyzed for mouse behavior using a Daniovision video tracking system (Noldus, Wageningen, The Netherlands). The speed and total movement distances (cm) were calculated and analyzed using the Etho Vision XT software (Noldus). We performed three independent experiments and averaged three values. 

### 2.8. Statistical Analysis

Values are presented as the mean ± standard deviation (SD) of three or more experiments. Data were analyzed using a nonparametric *t*-test in GraphPad Prism 6.0. All graphs were created using the GraphPad Prism version 6. Data are presented as mean ± SD and considered significant if *p* ≤ 0.05 (* *p* < 0.05, ** *p* < 0.005).

## 3. Results 

### 3.1. Synthesis of ADG-2e

As we demonstrated previously [36], 3′ and 5′ positions of the sugar in 3′-azidothymidine were functionalized to incorporate hydrophobic residues including alkyl and alkylaryl functionalities through an amide bond. To achieve cationicity, norspermidine and its functionalized guanidine were incorporated at the C4 position of a pyrimidine to mimic lysine and arginine, respectively [24]. The amphipathicity of the most potent compound ADG-2e was achieved by incorporating adamantane and guanidine as hydrophobic and hydrophilic counterparts, respectively. We observed that the guanidine mimetic (ADG-2e) was more effective than lysine mimetics (ADL-2e) owing to the distribution of cationic charge.

ADG-2e was synthesized as delineated in Figure 1. Briefly, the azidation of alcohol (1) was effected using tosyl chloride and sodium azide. To functionalize the C4 carbonyl of pyrimidine, 1,2,4-triazole was anchored in the presence of POCl_3_. Furthermore, K_2_CO_3_ mediated substitution reaction of diboc-norspermidine on 3 by replacing the triazole in acetonitrile resulted in the formation of 5 in 87% yield. To reduce the azide into amine for the incorporation of adamantyl group, Pd-C/H_2_, mediated reduction was performed, and subsequent coupling with 1-adamantaneacetic acid in presence of EDC, HOBt, and DIEA afforded the diamide, 6 in 86% yield. Further, deprotection of Boc groups on 6 was achieved by 1.25 M HCl to result in the formation of amine compound 7, which was under bisguanidinylation using 1*H*-pyrazole-1-carboxamidine hydrochloride and subsequent treatment of 1.25 M HCl solution in methanol resulted in the formation of ADG-2e in good yield [36].

### 3.2. Anticancer Effect of ADG-2e

After the successful synthesis of ADG-2e, its cytotoxicity was assessed against cancer cell lines, including HeLa CCL2 and BT549 as shown in Figure 1A. Interestingly, ADG-2e exhibited significant anti-proliferative effects against both HeLa CCL2 and BT549 cells. However, evaluation of ADG-2e against normal cells, including CCD34SK, CCD986SK, and MRC5 did not show considerable toxicity, demonstrating the selectivity over cancer cells. It is pertinent to note that ADG-2e inhibits cell growth at relatively low IC_50_ (from 4.8–9.7 μM) in cancer cell lines. 

To verify the induction of apoptosis, we examined ADG-2e against HeLa CCL2 cells using a live/dead viability/cytotoxicity kit that displays live and dead cells in green and red colors, respectively. As shown in Figure 1B, control group entirely consist of living cells marked with green signals. However, cells treated with 25 μM ADG-2e displayed a slight increase in dead cell counts. Furthermore, flow cytometric analysis of cell cycle distribution confirmed that treatment of 25 μM and 50 μM of ADG-2e on HeLa cells showed a small increase in dead cells by 8.8% and 13.4%, respectively (Figure 1C). These results speculate that apoptosis may not be the cause of cell death.

### 3.3. Effect of ADG-2e on the Cell Membrane 

To examine the induction of necrosis or apoptosis by ADG-2e, it is important to analyze the morphological changes and biochemical indicators of apoptosis in cancer cells. ADG-2e is an amphiphilic compound that can bind to cell membranes through its cationic components by interacting with the negatively charged phospholipids in the plasma membrane. As expected, the microscope images of ADG-2e treated HeLa CCL2 showed significant morphological changes on the membrane surface (Figure 2A). To accurately determine the effect of ADG-2e on cancer cell membranes, live-cell imaging experiments were performed, where the microscope (Carl Zeiss, Oberkochen, Germany) captured the images for 3 h at every 12 min interval [45]. Strikingly, cancer cells treated with ADG-2e showed significant changes in the cell membrane due to severe membrane shrinkage or burst, as indicated by red arrows (Figure 2B and Appendix A). Therefore, it is speculated that ADG-2e can affect the cytoplasmic proteins.

Furthermore, we investigated the effect of ADG-2e on nuclear protein lamin B, cytosolic proteins, including *α*-tubulin and *β*-actin, and apoptosis indicator proteins, namely caspase-3 and caspase-9. Western blot analysis (Figure 2C) revealed that *α*-tubulin and *β*-actin decreased in a dose-dependent manner when cells were treated with 50 μM ADG-2e. However, the nuclear protein, lamin B did not show any effect on ADG-2e treatment. This result was in agreement with the microscopic fluorescence images of HeLa CCL2 cells treated with ADG-2e (Figure 3), which showed that the nucleus was unaffected despite the fact that the membrane was disrupted. In general, the active form of caspase proteins is cleaved during cell death via apoptosis. However, ADG-2e treatment did not activate the apoptotic marker proteins, either caspase-3 or caspase-9, confirming that the induction of cell death was not facilitated by apoptosis (Figure 2C).

Generally, necrosis commences with cell swelling, which results in cell membrane rupture, and then release the cytoplasmic contents [46]. Necrotic cell death is morphologically characterized by cell swelling, which is known as oncosis [47]. As shown in Figure 2A,B, ADG-2e induced ruptures on plasma membrane and eventually disrupted membrane integrity, which led to subsequent leakage of intracellular contents such as *α*-tubulin and *β*-actin, suggesting tumor cell death through necrosis or oncosis pathway. To ascertain this, variation in the size of ADG-2e treated cells was analyzed using FACS. ADG-2e treated cells showed a dose-dependent decrease in cell size, indicating that the plasma membrane of the cells was destroyed by necrosis or oncosis (Figure 4). Collectively, these factors suggested that cell death presumably proceeded through necrosis or oncosis.

### 3.4. Evaluation of In Vivo Potency of ADG-2e

Inspired by the in vitro anticancer potency, ADG-2e was investigated for its chemotherapeutic ability in mice bearing HeLa cell xenograft tumors, which were subjected to IP injections of ADG-2e or vehicle three times a week, as scheduled in Figure 5A. The cancer volume growth of each mouse was measured and compared at every three days interval. In contrast to the vehicle, ADG-2e treated mice group showed remarkable tumor growth suppression after the final injection, which was approximately 50% volume reduction (Figure 5B–D).

To ascertain whether the reduction in tumor size was entirely due to the anti-tumor effect of ADG-2e, we investigated the change in body weight after every injection, as shown in Figure 6A. The results suggested that the body weight of mice did not show any changes, proving the anti-tumorigenic effect of ADG-2e. Furthermore, variations in mouse movements after twenty-four hours of the final injection were investigated (Figure 6B) by analyzing the movement distances using the Etho Vision XT software. The results suggest that ADG-2e treated mice showed considerable improvements in the movement travelled compared to vehicle-treated mice. Therefore, body weight change and activity change experiments demonstrate the anti-tumor potential of ADG-2e. To verify that anti-tumorigenesis proceeded through apoptosis, cancer tissues were investigated for cleavage of caspase 3 signal. As shown in Figure 7, the immunochemical data did not show any signals for caspase 3 cleavage, suggesting that anti-tumorigenesis of ADG-2e probably proceeded through the necrosis or oncosis pathway.

## 4. Discussion and Conclusions

Cancer is a heterogeneous disorder with a steep increase in incidence and mortality rates, and remains a threat to socioeconomic life worldwide [1]. Although various treatments including surgery, radiation, and chemotherapy have been employed, the survival rate of cancer patients has not improved considerably because of the migration of cancer cells from the primary site to distant organs through metastasis [12]. At present, over 200 antitumor drugs exist in clinical treatments, but very few of them inhibit metastasis [39]. Therefore, the development of a potential anticancer agent with antimetastatic potential is necessary to achieve successful therapeutic outcomes.

A recently reported amphipathic small molecule, ADG-2e was found to display significant potency against tumor metastasis. To address the above-mentioned issues and to evaluate the anticancer effects of ADG-2e, we investigated its anticancer potential. Interestingly, ADG-2e showed a significant anti-proliferative effect, both in vitro and in vivo. ADG-2e inhibited the proliferation of HeLa CCL2 and BT549 cancer cells, whereas normal cells were not affected, indicating selective anti-cancer effects. Moreover, ADG-2e showed a significant anti-tumorigenic effect as a result of cancer inhibition in a cancer-bearing mouse xenograft model. The selective anticancer effect of ADG-2e might be attributed to the fact that the existence of electrostatic binding between the positive charges of ADG-2e and the negative charges located at the outer part of the cancer cell membranes. However, such electrostatic binding is not feasible in normal cells because the negatively charged phosphatidylserine localizes in the inner leaflet of the membrane, whereas the outer leaflet of the plasma membrane comprises neutral phospholipids, including phosphatidylcholine and sphingomyelins [8,9]. Furthermore, in animal models, ADG-2e exhibited anti-cancer effects without causing any changes in body weight. After the treatment of the inhibitor, mice were found to increase mobility, suggesting the potential of ADG-2 to develop as a model for developing anti-cancer drugs. 

Anticancer peptides (ACP) were reported to facilitate cell death through various processes, including necrosis due to cell membrane lysis, modulation of immune responses, inhibition of kinase, interference with functional proteins, and induction of apoptosis [17]. Apoptosis and accidental cell death are the key modalities of cancer cell death that have been reported [39,40]. Apoptosis is a programmed cellular suicide of unhealthy cells that involves two different pathways namely, (i) extrinsic pathway by activating caspases, (ii) intrinsic pathway through mitochondrial outer membrane permeabilization (MOMP), which induces proapoptotic proteins [33,48]. We observed that ADG-2e treated cancer cells showed damaged plasma membrane. Consequently, swelling of the cell membrane was also observed, which is characteristic of oncosis. Usually, oncosis leads to necrosis, which is associated with cell shrinkage and karyolysis. Thus, necrosis contradicts the apoptosis [47]. Indeed, anti-cancer agents such as kahalalide F [49] and artesunate [50] induce cell death through oncosis. Morphological changes and edema in the cancer cell membrane suggest the potential of ADG-2e to evolve as an anti-cancer agent, following the necrotic cell death pathway. In summary, together these results suggest that ADG-2e has the potential to evolve as a therapeutic model for treating cancer. 

## Data Availability

All data available are reported in the article.

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
