# Peer review of "Amphipathic Small Molecule AZT Compound Displays Potent Inhibitory Effects in Cancer Cell Proliferation"

_pharmaceutics, 2021, doi:10.3390/pharmaceutics13122071_

Round 1

Reviewer 1 Report

Thank you for inviting me to evaluate the article titled “Amphipathic small molecule AZT compound displays potent inhibitory effects in cancer cell proliferation”. The manuscript described the fabrication of AZT based small molecule (ADG-2e) that revealed a notable potency against tumor metastasis. Therefore, ADG-2e can potentially evolve as an anticancer and antimetastatic agent and which may represent a model for nucleoside-based small molecule anticancer drug discovery. I recommend a minor revision before accepted, and I have the following comments for the authors to further improve the article.

  1. Line 228, what was evaluated? Please make a correction to the expression of this sentence.
  2. The image clarity of Figure 1B is too low, and the fluorescent staining is very unclear. Please replace the image with a clearer one.
  3. Line 274, what’s the meaning of “which often may be overlooked and misinterpretation is also likely”? Please make a careful revision of the grammar and expressions for there are too many errors in this paper and the language in the article is very lacking.
  4. Line 310, “and the release the cytoplasmic contents” should be “and then release the cytoplasmic contents”.
  5. The meaning of the numerical symbols on the left side of Figure 5C is not clearly explained in the manuscript.
  6. The image clarity of Figure 7 is also too low, please change a clearer one.
  7. Since the immunochemical data did not show any signals for Caspase 3 cleavage, how to determine the pathway of tumor death.

Author Response

Point by point response to the reviewer is attached below

Reviewer 2 Report

The author synthesized a small molecule (ADG-2e) based on 3¢-azido-3¢-deoxythymidine (AZT). This compound showed significant inhibition of cancer cell proliferation on HeLa CCL2, cervical cancer cells and BT549 breast ductal carcinoma in vitro. Finally, the anti-tumor potential of ADG-2e in a mouse xenograft model was investigated. This research work is innovative, but the author also needs to deal with the following questions:

  1. In the discussion, the author said “the potential anticancer effect of ADG-2e might be attributed to the fact that the existence of electrostatic binding between the positive charges of ADG-2e and the negative charges of the cancer cell membranes”. For normal cells, its membrane is also negatively charged, so why doesn't it inhibit normal cells? Please explain it.
  2. The author said the compound can inhibit tumor metastasis. It is best for the author to establish a mouse metastasis model and evaluate its anti-metastatic ability in vivo.
  3. The author should add the pharmacokinetic data of the drug.
  4. As a new type of compound, the author should elaborate on the physical and chemical properties of the compound, such as hydrophilic and lipophilic value, crystal form, etc.
  5. Why did the experiment take intraperitoneal injection instead of oral or intravenous administration?
  6. What is the maximum tolerated dose of this compound in mice?
  7. Why is the amount of the last beta-action different from the other groups in Fig.2c?
  8. Please add statistical differences for Fig.1a.
  9. Please check the statistical results and expressions of fig 5D. In the legend, the author writes *P>0.001. Please give the specific value of this one.

Author Response

Response to the reviewer 

  1. In the discussion, the author said “the potential anticancer effect of ADG-2e might be attributed to the fact that the existence of electrostatic binding between the positive charges of ADG-2e and the negative charges of the cancer cell membranes”. For normal cells, its membrane is also negatively charged, so why doesn't it inhibit normal cells? Please explain it.

       Response: Thank you, the explanation is incorporated in the discussion part         of manuscript as given below

”The selective anticancer effect of ADG-2e might be attributed to the fact that the existence of electrostatic binding between the positive charges of ADG-2e and the negative charges located at the outer part of the cancer cell membranes. However, such electrostatic binding is not feasible in normal cells because the negatively charged phosphatidylserine localizes in the inner leaflet of the membrane, whereas the outer leaflet of the plasma membrane comprises neutral phospholipids, including phosphatidylcholine and sphingomyelins [8,9].”

  1. The author said the compound can inhibit tumor metastasis. It is best for the author to establish a mouse metastasis model and evaluate its anti-metastatic ability in vivo.

Response: Thank you for the valuable suggestion, we already started in vivo tumor metastasis and pharmacokinetics experiments, and detailed studies are under progress, which will be published separately in the near future.

  1. The author should add the pharmacokinetic data of the drug.

Response: As we stated in question 2, detailed study on in vivo tumor metastasis and pharmacokinetics are under progress, which will be published separately in the near future.

  1. As a new type of compound, the author should elaborate on the physical and chemical properties of the compound, such as hydrophilic and lipophilic value, crystal form, etc.

Response: thank you, physical and chemical properties are attached at the bottom of the compound description in the experimental section given in supporting information.

  1. Why did the experiment take intraperitoneal injection instead of oral or intravenous administration?

Response: We performed the mice in vivo experiment in moderate conditions because we did not have any mice PO (oral administration) or IV pilot data.

  1. What is the maximum tolerated dose of this compound in mice?

Response: Please find the “Reviewer only-02.file“, which summarizes Balb/c mouse toxicity pilot test with a table.

The experiments suggest that mice in vivo experiment was only available in 25 mg/kg (IP) but not in 50mg/kg (IP).

  1. Why is the amount of the last beta-actin different from the other groups in Fig.2c?

Response: After ADG-2e treatment, we selected cytoplasmic proteins such as beta-actin and alpha-tubulin to account for cytoplasmic protein leakage. In this experiment, neither of them didn’t serve as a loading control.

  1. Please add statistical differences for Fig.1a.

Response: As recommended by the reviewer, we added the statistical evaluation.

  1. Please check the statistical results and expressions of fig 5D. In the legend, the author writes *P>0.001. Please give the specific value of this one.

Response: As recommended by the reviewer, we corrected the statistical evaluation in the Figure 5D graph.

Reviewer 3 Report

The authors report a study on an amphipathic small molecule that displays potent  inhibitory effects in cancer cell proliferation. The topic is interesting and the conclusions are sound. I think that tha paper can be accepted in the present form.

Author Response

We thank the reviewer for accepting our manuscript.